Measurement of dispersion of PM 2.5 in Thailand using confidence intervals for the coefficient of variation of an inverse Gaussian distribution

Chankham Wasana
Niwitpong Sa-Aat sa-aat.n@sci.kmutnb.ac.th
Niwitpong Suparat
Department of Applied Statistics, King Mongkut’s University of Technology North Bangkok , Bangkok , Thailand
Chai Tianfeng
Electronic publication date: 2022 Feb 17
Publication date: 2022
Volume: 10
Electronic Location ID: e12988
Received 2021 Oct 22; Accepted 2022 Feb 1
Copyright: ©2022 Chankham et al.
Copyright year: 2022
Copyright holder: Chankham et al.
License: This is an open access article distributed under the terms of the Creative Commons Attribution License, which permits unrestricted use, distribution, reproduction and adaptation in any medium and for any purpose provided that it is properly attributed. For attribution, the original author(s), title, publication source (PeerJ) and either DOI or URL of the article must be cited.
License URL: https://creativecommons.org/licenses/by/4.0/

Keywords: Air pollution, Particulate matter, Pollution data, Adjusted generalized confidence interval, Bootstrap percentile confidence interval, Fiducial confidence interval, Fiducial highest posterior

Funding: King Mongkut’s University of Technology North Bangkok KMUTNB-PHD-63-01 This research was funded by King Mongkut’s University of Technology North Bangkok, contract number: KMUTNB-PHD-63-01. The funders had no role in study design, data collection and analysis, decision to publish, or preparation of the manuscript.

==============================
Air pollution is a growing concern for the general public in Thailand with PM 2.5 (particulate matter ≤ 2.5 µm) having the greatest impact on health. The inverse Gaussian (IG) distribution is used for examining the frequency of high concentration events and has often been applied to analyze pollution data, with the coefficient of variation (CV) being used to calculate the quantitative difference in PM 2.5 concentrations. Herein, we propose confidence intervals for the CV of an IG distribution based on the generalized confidence interval (GCI), the adjusted generalized confidence interval (AGCI), the bootstrap percentile confidence interval (BPCI), the fiducial confidence interval (FCI), and the fiducial highest posterior density confidence interval (F-HPDCI). The performance of the proposed confidence intervals was evaluated by using their coverage probabilities and average lengths from various scenarios via Monte Carlo simulations. The simulation results indicate that the coverage probabilities of the AGCI and FCI methods were higher than or close to the nominal level in all of test case scenarios. Moreover, FCI outperformed the others for small sample sizes by achieving the shortest average length. The efficacies of the confidence intervals were demonstrated by using PM 2.5 data from the Din Daeng and Bang Khun Thian districts in Bangkok, Thailand.

Introduction

Air pollution is regarded as a serious environmental threat. Inefficient forms of transportation (polluting fuels and cars), coal-fired power plants, and agricultural and garbage burning are all major causes of air pollution, high levels of which have been linked with several negative health effects (Pope & Dockery, 2006). Short and long-term exposure to air pollution has been linked to negative health outcomes, and respiratory infections, heart problems, and lung cancer are all increased by it, with people who are already sick being subjected to more severe consequences, while children and the elderly are particularly vulnerable. One of the most health-damaging pollutants is PM 2.5 (particulate matter ≤2.5 µm), which can penetrate deep into the lungs.

The problem of air pollution is a growing concern for the general public in Thailand. Especially, Bangkok’s urban environment is a complex mixture of commercial, residential, and industrial buildings. The growing number of vehicles on the road and increased energy consumption constitute a significant source of PM 2.5 emissions, and consequently, PM 2.5 levels (and population exposure) along roadsides are frequently substantially greater than in other areas. Industrial emissions are also of great concern in some areas. Chuersuwan et al. (2008) investigated PM 10 (particulate matter ≤10 µm) and PM 2.5 concentrations in the Bangkok Metropolitan Region over the course of a year and uncovered their key sources in the inner city adjacent to a busy road. In the Bangkok Metropolitan Region, Oanh et al. (2013) reported the mass concentrations of PM 2.5 and gaseous pollutants along specific travel routes and at fixed roadside locations. Since the coefficient of variation (CV) was used in these studies to quantitatively characterize the temporal variation of PM 2.5, we are interested in analyzing PM 2.5 data by using the CV of an IG distribution.

The inverse Gaussian (IG) distribution is an excellent choice for modeling positive and right-skewed data, and some of its statistical features were discussed by Tweedie (1957). The IG distribution has been used in hydrology, cardiology, pharmacokinetics, demography, economics, finance, among others. Schrodinger (1915) used it to model the first passage of the time distribution for Brownian motion. Chhikara & Folks (1989) proposed its application to model lifetime and wind energy data. The IG distribution has also been used to investigate insulating fluid failure time, market incidence models, particle cycle time distribution in the blood, health costs, and air pollution. For example, Kumar & Goel (2016) calculated PM 10, PM 2.5, and PM 1 zones of influence under various driving situations and fitted the pdfs of fixed-site data for different PM types at signalized traffic junctions, for which the inverse Gaussian was found to be the most suitable. Gavril et al. (2006) used the pdf in the analysis of the distribution of air pollution in central Athens; they found that the inverse Gaussian distribution provided a better result than the beta, gamma, and Weibull distributions. Therefore, we are interested in studying PM 2.5 pollution by using the IG distribution.

Numerous investigations on the parameters and confidence intervals for IG distributions have previously been conducted. For example, Arefi, Borzadaran & Vaghei (2008) proposed the likelihood ratio interval and the Wald interval for calculating the mean of an IG distribution. Ye, Ma & Wang (2010) employed the generalized confidence interval (GCI) approach for hypothesis testing and interval estimation for the common mean of several IG populations. Tian & Wilding (2005) presented confidence intervals of the ratio of means of two independent IG populations using modified directed likelihood ratio statistics. Krishnamoorthy & Tian (2008) proposed a method for constructing confidence intervals for IG based on the generalized variable approach and a modified likelihood ratio test. In another study, Ismail & Auda (2007) used Bayesian and fiducial inference via the Gibbs sampling process for IG distributions with Type-II censored data. Jayalath & Chhikara (2020) developed a thorough survival analysis for an IG distribution via the Gibbs sampling method by employing Bayesian and fiducial approaches that require a Monte Carlo Markov Chain (MCMC) method.

The CV, which is the ratio of the standard deviation to the mean (it is useful to measure data dispersion with different units), has been used in a variety of fields, including agriculture, biology, medicine, and finance, with many researchers having developed CV parameters and confidence intervals. Banik & Kibria (2011) presented confidence intervals for estimating the CVs of symmetric and positively skewed distributions. Wongkhao, Niwitpong & Niwitpong (2015) suggested confidence intervals for the CV of a normal distribution. Sangnawakij & Niwitpong (2017) used three approaches: the method of variance of estimates recovery (MOVER), GCI, and the asymptotic confidence interval to establish confidence intervals for the CV of a two-parameter exponential distribution. Niwitpong (2013) created a new confidence interval for the CV of a lognormal distribution with restricted parameters. Yosboonruang, Niwitpong & Niwitpong (2019) provided the confidence intervals for the CV of a lognormal distribution by utilizing Bayesian and fiducial GCI methods. Confidence intervals for the difference between the CVs of Weibull distributions were proposed by La-ongkaew, Niwitpong & Niwitpong (2021). Chankham, Niwitpong & Niwitpong (2019) used GCI and bootstrap percentile confidence interval (BPCI) methods to calculate confidence intervals for the CV of an IG distribution.

The purpose of the current study is to establish new confidence intervals for the CV of an IG distribution by using generalized confidence interval (GCI), adjusted generalized confidence interval (AGCI), bootstrap percentile confidence interval (BPCI), fiducial confidence interval (FCI), and fiducial highest posterior density confidence interval (F-HPDCI) methods. The paper is organized as follows. The ideas of the proposed methods are clarified in Methods section. The simulation results are presented in Results section, and then the proposed methods are applied to the real world datasets, as detailed in An empirical application. The last two sections contain discussion and conclusions on the study.

Methods

Let X = (X1, X2, …, Xn) be a random sample from an IG distribution denoted as IG (μ, λ), where μ and λ are the mean and scale parameters of X, respectively. Subsequently, the probability density function of X is given by (1) fx,μ,λ=λ2πx312 exp−λx−μ22μ2x,x>0,μ>0,λ>0,

The maximum likelihood estimates (MLEs) of parameters μ and λ are μ=X ¯=1n∑i=1niXi and λi−1=1ni∑j=1niXij−1−X ¯i−1, respectively. For notation convenience, we use Vi=λi−1.Xi and Vi are mutually independent random variables with respective distributions (2) X¯i∼IGμ,niλiandniλiVi∼χni−12,

where χni−12 denotes a chi-square distribution with ni − 1 degrees of freedom. Reproducing the exponential property of the IG explains the independence of these two statistics. It is simple to show that X ¯i,Vi are form a set of complete sufficient statistics. The population mean and variance of X can respectively be expressed as (3) EX=μ,

and (4) VarX=μ3λ.

Therefore, the CV of X is (5) CVX=θ=VarXEX=μλ.

Here, we present the five methods for constructing confidence intervals for θ.

The GCI method

Since the GCI approach was first introduced by Weerahandi (1993), several researchers have used it to provide statistical inferences (Tsui & Weerahandi, 1989; Weerahandi, 1993; Weerahandi, 1995; Ye & Wang, 2007; Krishnamoorthy & Tian, 2008). After that, Ye, Ma & Wang (2010) presented the generalized pivot quantity (GPQ) criterion for the parameters and the constructed confidence intervals for the common mean of several inverse Gaussian populations. Furthermore, Chankham, Niwitpong & Niwitpong (2019) recommended GCI for constructing confidence intervals for the coefficient of variation of an IG distribution; they found that GCI provide coverage probabilities greater than or equal to the nominal confidence level at 0.95. Therefore, GCI was selected as the baseline for comparison with the proposed methods of this study.

The confidence interval for the CV is calculated using the concept of the GPQ. Let X = (X1, X2, …, Xn) be random variables from a distribution defined by probability density function fX(x; θ, δ), with θ and δ being the sought after and nuisance parameters, respectively. Meanwhile, GPQ R(X; x, θ, δ) satisfies the following two conditions.

(i) The probability distribution of function R(X; x, θ, δ) is independent of the unknown parameters.

(ii) The observed value of R(X; x, θ, δ), X = x does not depend on the nuisance parameters.

If R(X; x, θ, δ) satisfies both conditions, then the GCI for the parameter of interest is calculated using the percentiles of the GPQ. Let [Rα/2, R1−α/2] be a 100(1 − α)% two-sided GCI for the parameter of interest, where Rα and R1−α are denoted as 100(α/2)% and 100(1 − α/2)% of R(X; x, θ, δ), respectively.

Ye, Ma & Wang (2010) proposed the respective GPQs for λi and μi as follows: (6) Rλi=niλiViniυi∼χni−12niυi,i=1,2,…,k,

where χni−12 denotes a chi-squared distribution with ni − 1 degrees of freedom. Thus, the GPQ for μi is given by (7) Rμi=x¯i|1+niλix¯i−μμx¯ix¯iniRλi|∼dxi ¯1+Zix¯iniRλi,

where ∼d is approximately distributed and Zi ∼ N(0, 1). The approximate distribution is derives from the moment matching method of Chhikara & Folks (1989). Note that niλiX¯−μi/μiX¯i is a limiting distribution of N(0, 1). Consequently, the observed value of Rμi is μi. Therefore, the GCI for the CV of an IG distribution is given by (8) Rθ=RμRλ.

The 100(1 − α)% two-sided confidence interval for the CV of an IG distribution based on the GCI method is given by (9) CIGCI=LGCI,UGCI=Rθα/2,Rθ1−α/2,

where Rθ(α/2) and Rθ(1 − α/2) are the 100(α/2)-th and 100(1 − α/2)-th percentiles of the distribution of Rθ, respectively.

The following algorithm was used to construct GCI:

Algorithm 1

(1) Generate X1, X2, …, Xn from an IG distribution.

(2) Compute μ ˆ and λ ˆ.

(3) Generate χn−12 from a Chi-square distribution with n − 1 degrees of freedom and Z from a standard normal distribution.

(4) Use Eqs. (6), (7) and (8) to calculate Rλ, Rμ,  and Rθ, respectively.

(5) Repeat Steps 3-4, 5,000 times and obtain an array of Rθ.

(6) Calculate the 95% confidence intervals for θ by using Eq. (9). If L ≤ θ ≤ U, then set cp = 1 ; else , set cp = 0.

(7) Repeat Steps 1-6, 15,000 times to compute the coverage probability and the average length.

The AGCI method

According to Ye, Ma & Wang (2010), an approach similar to the GCI method can be utilized for the single coefficient of variation. The GPQ of λ uses the same of GCI method. Subsequently, Krishnamoorthy & Tian (2008) established an estimated GPQ of μ ~i as follows: (10) Rμi ~=X ¯imax0,1+tni−1X ¯ivini−1,

where tni−1 denotes a t-distribution with tni−1 degrees of freedom. However, the denominator can become zero when tni−1 takes a negative value, and thus Rμ ~i is an approximate GPQ.

Therefore, the AGCI for the CV of an IG distribution is given by (11) Rθ ~=Rμ ~Rλ.

Subsequently, the 100(1 − α)% two-sided confidence interval for the CV of an IG distribution based on the AGCI method is given by (12) CIAGCI=LAGCI,UAGCI=Rθ ~α/2,Rθ ~1−α/2,

where Rθ ~α/2 and Rθ ~1−α/2 which are the 100(α/2)-th and 100(1 − α/2)-th percentiles of the distribution of Rθ ~, respectively can be obtained from the notion of Algorithm 1.

The BPCI method

When applying this method, the distribution of the bootstrap sample statistic is a direct approximation of the data sample (Efron & Tibshirani, 1986). It is proceeded by re-sampling the data with replacement from the distribution. Chankham, Niwitpong & Niwitpong (2019) reported that BPCI performed more poorly than GCI. However, to provide context, this approach was still included in the comparative analysis. Suppose X = (X1, X2, …, Xn) is a random sample of size n from an IG distribution. Sampling is replaced by X∗=X1∗,X2∗,…,Xn∗, which can be obtained by the bootstrapping the sample B times. Efron & Tibshirani (1986) claimed that a minimum of approximately B = 1, 000 bootstrap resamples is usually sufficient for obtaining reasonable accurate confidence interval estimates for CV of IG distribution.

The 100(1 − α)% two-sided confidence interval for the CV of an IG distribution based on BPCI is defined as (13) CIBPCI=LBPCI,UBPCI=θ∗α/2,θ∗1−α/2,

where θ∗(α/2) and θ∗(1 − α/2) are percentiles of the distribution.

The following algorithm is used to construct BPCI:

Algorithm 2

(1) Generate X1, X2, …, Xn from an IG distribution.

(2) Obtain bootstrap sample X1∗,X2∗,…,Xn∗ from Step 1.

(3) Compute θ∗.

(4) Repeat Steps 2 and 3, 1,000 times.

(5) Compute 95% confidence interval based on BPCI according to Eq. (13).

The FCI method

Although fiducial inference proposed by Fisher (1973) is similar to the Bayesian framework, it does not require prior knowledge of the distribution for estimating the parameters involved. Fiducial inference is the only type of inference with frequentist interpretation that uses conditionality on the data. Hence, it allows the implementation of Gibbs sampling (Geman & Geman, 1984), which is an MCMC method commonly used to generate a sample from the posterior distribution for Bayesian inference by sweeping through a variable to sample from its conditional distribution while the remaining variables are fixed at their current values. The sampling distributions of the MLEs of both μ and λ are used for an IG distribution. The fiducial distributions of μ and λ are easily obtained simply by replacing them with their MLEs when they appear in their sample distributions as follows (14) μ∼IGμ ˆ,nλ ˆ,

and (15) λ∼λ ˆ/nχn−12,

where μ ˆ and λ ˆ are the MLEs of the μ and λ.

Subsequently, the 100(1 − α)% two-sided confidence interval for the CV of an IG distribution based on fiducial inference is given by (16) CIFCI=LFCI,UFCI=θtα/2,θt1−α/2,

where θt(α/2) and θt(1 − α/2) denotes the 100(α/2) -th and 100(1 − α/2) -th percentiles of θt, respectively.

Algorithm 3 The Gibbs sampling

(1) Take the initial values (MLEs) of parameters (μ(0), λ(0))

(2) Generate μt∼IGμ ˆt−1,nλ ˆt−1

(3) Generate λt∼λ ˆt−1/nχn−12

(4) Repeat Step 2–3, T times, where T is the number of MCMC replications.

(5) Burn-in 1,000 samples and compute the parameter of interest.

(6) Compute the 95% confidence interval based on the fiducial inference Eq. (16).

The F-HPDCI method

Hear, the HPD credible interval of the parameters are obtained by using the MCMC method (Chen & Shao, 1999). It is assumed that each value inside the interval has a higher posterior density than any of the values outside of it (Box & Tiao, 2011).

Hence, the 100(1 − α)% two-sided confidence interval for the CV of an IG distribution based on the F-HPDCI is given by (17) CIF−HPDCI=LF−HPDCI,UF−HPDCI.

The following algorithm is used to construct confidence interval using the F-HPDCI method for the CV of an IG distribution:

Algorithm 4

(1) Take the initial values (MLEs) of parameters (μ(0), λ(0))

(2) Use algorithm 3 to calculate the parameter of interest.

(3) Compute the 95% confidence interval based on F-HPDCI according to Eq. (17).

RESULTS

A Monte Carlo simulation study using the R statistical programming language was conducted to evaluate the performances of the confidence intervals based on GCI, AGCI, BPCI, FCI, and F-HPDCI for the CV of an IG distribution. The sample size was set as n = 5, 10, 30, 50, 100,  and 200; μ as 0.5 and 1, and λ as 1, 2, 5, and 10. We used 15,000 replications for each parameter combination. Furthermore, 5,000 repetitions were used for the GCI and AGCI method, 1,000 bootstrap samples for the BPCI method, and 20,000 realizations of MCMC using the Gibbs algorithm with a burn-in of 1,000 for the fiducial methods. Assessing the performances of the confidence intervals of the five methods was achieved in terms of their coverage probabilities and the average lengths respectively calculated as (18) CP=cL≤θ≤UM,

and (19) AL=∑i=1MU−LM.

where c(L ≤ θ ≤ U) is the numbers of simulation replications for θ that lie within the confidence interval. The best-performing confidence interval in each case had a coverage probability is greater than or equal to the nominal confidence level of 0.95 and the shortest average length. M is the number of simulation replications. The computational steps to estimate the coverage probabilities and average length performances of all of the methods were computed by using Algorithm 5.

Algorithm 5.

(1) Set the values M, m, n, μ, and λ.

(2) Generate X1, X2, …, Xn from an IG distribution.

(3) Use Algorithm 1 to construct GCI [LGCI, UGCI] and AGCI [LAGCI, UAGCI].

(4) Use Algorithm 2 to construct BPCI [LBPCI, UBPCI].

(5) Use Algorithm 3 to construct FCI [LFCI, UFCI].

(6) Use Algorithm 4 to construct F-HPDCI [LF−HPDCI, UF−HPDCI].

(7) If (L ≤ θ ≤ U) , set P = 1; else set P = 0.

(8) Calculate (U–L).

(9) Repeat Steps 2–8, M times.

(10) Compute the coverage probabilities and the average lengths.

The results in Table 1, Figs. 1 and 2 show that GCI, AGCI, and FCI provided coverage probabilities greater than or close to the nominal coverage level of 0.95 in almost all cases. However, those of F-HPDCI were less than the nominal confidence level for some cases (n = 5, μ = 0.5, 1, λ = 1, 2; n = 10, μ = 0.5, 1, λ = 1; and n = 30, μ = 1, λ = 1) and those of BPCI in all cases. Meanwhile, the average lengths of all methods became narrower when the sample size was increased, with FCI and F-HPDCI methods providing the shortest ones for n = 5, 10, or 30 and AGCI for n = 50, 100, or 200.

Table 1 The coverage probabilities (CP) and average length (AL) of 95% two-sided confidence intervals for the coefficient of variation of inverse Gaussian distribution.

n	μ	λ	CP	AL	
			GCI	AGCI	BPCI	FCI	F-HPDCI	GCI	AGCI	BPCI	FCI	F-HPDCI	
5	0.5	1	0.9750	0.9466	0.5351	0.9647	0.9373	2.622	1.514	0.635	1.564	1.322	
		2	0.9638	0.9489	0.5342	0.9625	0.9478	1.526	1.071	0.443	1.092	0.921	
		5	0.9568	0.9471	0.5447	0.9523	0.9474	0.782	0.676	0.276	0.681	0.573	
		10	0.9534	0.9482	0.5506	0.9520	0.9470	0.515	0.481	0.195	0.483	0.406	
	1	1	0.9809	0.9483	0.5057	0.9579	0.9235	4.234	2.127	0.913	2.261	1.909	
		2	0.9741	0.9478	0.5335	0.9640	0.9382	2.609	1.511	0.634	1.566	1.323	
		5	0.9690	0.9536	0.5348	0.9649	0.9524	1.281	0.957	0.392	0.970	0.817	
		10	0.9623	0.9547	0.5480	0.9592	0.9521	0.786	0.679	0.277	0.684	0.576	
10	0.5	1	0.9781	0.9490	0.7076	0.9591	0.9432	1.118	0.783	0.534	0.841	0.780	
		2	0.9696	0.9492	0.7203	0.9592	0.9502	0.660	0.555	0.367	0.577	0.535	
		5	0.9598	0.9501	0.7248	0.9529	0.9519	0.377	0.352	0.229	0.357	0.331	
		10	0.9568	0.9525	0.7210	0.9538	0.9528	0.256	0.248	0.159	0.250	0.231	
	1	1	0.9892	0.9531	0.6986	0.9566	0.9306	2.208	1.105	0.792	1.256	1.166	
		2	0.9806	0.9486	0.7103	0.9614	0.9477	1.121	0.784	0.537	0.843	0.783	
		5	0.9673	0.9509	0.7235	0.9571	0.9541	0.571	0.497	0.329	0.513	0.476	
		10	0.9588	0.9488	0.7221	0.9545	0.9510	0.375	0.350	0.227	0.356	0.330	
30	0.5	1	0.9789	0.9478	0.8652	0.9559	0.9480	0.484	0.384	0.361	0.425	0.414	
		2	0.9704	0.9498	0.8711	0.9552	0.9545	0.308	0.272	0.244	0.286	0.279	
		5	0.9606	0.9520	0.8649	0.9530	0.9506	0.181	0.172	0.150	0.176	0.171	
		10	0.9562	0.9502	0.8700	0.9520	0.9500	0.125	0.120	0.105	0.123	0.119	
	1	1	0.9934	0.9486	0.8555	0.9500	0.9423	0.832	0.545	0.549	0.652	0.637	
		2	0.9812	0.9487	0.8618	0.9551	0.9492	0.484	0.384	0.360	0.424	0.414	
		5	0.9652	0.9471	0.8617	0.9503	0.9479	0.125	0.243	0.216	0.253	0.247	
		10	0.9591	0.9500	0.8637	0.9503	0.9479	0.181	0.172	0.150	0.175	0.171	
50	0.5	1	0.9820	0.9501	0.8964	0.9505	0.9495	0.359	0.289	0.290	0.321	0.316	
		2	0.9730	0.9517	0.8948	0.9505	0.9495	0.229	0.204	0.195	0.216	0.213	
		5	0.9621	0.9520	0.8965	0.9531	0.9517	0.136	0.129	0.120	0.132	0.130	
		10	0.9559	0.9505	0.8922	0.9501	0.9498	0.094	0.091	0.084	0.092	0.091	
	1	1	0.9930	0.9530	0.8899	0.9549	0.9480	0.598	0.409	0.443	0.493	0.485	
		2	0.9837	0.9552	0.8956	0.9575	0.9521	0.359	0.289	0.289	0.320	0.315	
		5	0.9677	0.9514	0.8895	0.9515	0.9470	0.201	0.182	0.172	0.190	0.187	
		10	0.9602	0.9504	0.8973	0.9531	0.9529	0.136	0.129	0.120	0.132	0.130	
100	0.5	1	0.9856	0.9547	0.9236	0.9521	0.9500	0.247	0.200	0.212	0.223	0.221	
		2	0.9718	0.9519	0.9233	0.9516	0.9509	0.159	0.142	0.142	0.149	0.148	
		5	0.9634	0.9526	0.9246	0.9523	0.9519	0.094	0.089	0.087	0.092	0.091	
		10	0.9557	0.9506	0.9246	0.9523	0.9519	0.065	0.063	0.061	0.064	0.063	
	1	1	0.9931	0.9516	0.9129	0.9496	0.9460	0.407	0.283	0.324	0.344	0.341	
		2	0.9839	0.9531	0.9209	0.9528	0.9488	0.247	0.200	0.212	0.223	0.221	
		5	0.9696	0.9526	0.9222	0.9526	0.9509	0.139	0.127	0.126	0.133	0.131	
		10	0.9633	0.9521	0.9225	0.9530	0.9519	0.094	0.084	0.087	0.092	0.091	
200	0.5	1	0.9830	0.9521	0.9398	0.9551	0.9540	0.172	0.140	0.152	0.156	0.155	
		2	0.9736	0.9517	0.9325	0.9526	0.9506	0.111	0.099	0.102	0.105	0.104	
		5	0.9618	0.9519	0.9394	0.9535	0.9518	0.066	0.063	0.063	0.064	0.064	
		10	0.9557	0.9508	0.9373	0.9503	0.9483	0.045	0.044	0.044	0.045	0.045	
	1	1	0.9944	0.9535	0.9344	0.9521	0.9510	0.282	0.198	0.235	0.241	0.240	
		2	0.9847	0.9503	0.9346	0.9520	0.9500	0.172	0.140	0.152	0.156	0.156	
		5	0.9678	0.9522	0.9365	0.9505	0.9504	0.097	0.089	0.091	0.093	0.092	
		10	0.9565	0.9509	0.9348	0.9505	0.9499	0.066	0.063	0.063	0.064	0.064	
Notes.

GCI generalized confidence interval

AGCI adjusted generalized confidence interval

BPCI bootstrap percentile confidence interval

FCI fiducial confidence interval

F-HPDCI fiducial hightest posterior density confidence interval

Bold denoted as the coverage probability ⩾0.95 and the shortest average length.

Figure 1 Comparison of the performance of the proposed methods with sample size in terms of coverage probability.

Figure 2 Comparison of the performance of the proposed methods with sample size in terms of average length.

An Empirical Application

Example 1

PM 2.5 from the Din Daeng district of Bangkok were collected by the Pollution Control Department, Thailand (http://air4thai.pcd.go.th/webV2/download.php) due to this area having a high traffic volume (Table 2). Figure 3 exhibits a Q–Q plot of the PM 2.5 data, indicating that an IG distribution is suitable for this dataset. Before computing the confidence intervals, the minimum Akaike information criterion (AIC) and Bayesian information criterion (BIC) were first used to test the best-fitting distribution for these data. These two criteria are respectively defined as (20) AIC=−2lnL+2k,

and (21) BIC=−2lnL+2klnn,

where L is a likelihood function, k is the number of parameters, and n is the number of recorded measurements.

It was found that the PM 2.5 data fit an IG distribution because the AIC and BIC values for this distribution were smaller than the other tested distributions (normal, lognormal, Cauchy, exponential, and Weibull) (Table 3). The summary statistics were computed: n=31,μ ˆ=53.1229,λ ˆ=337.9519, and CV = 0.3965. The 95% confidence intervals for the CV of the PM 2.5 dataset by using the five methods are reported in Table 4.These results agree with the simulation results for n = 30 in that the F-HPDCI method performed the best in terms of coverage probability and average length, which supports the simulation results.

Example 2

PM 2.5 data from the Bang Khun Thian district, Bangkok, were collected by the pollution Control Department, Thailand (http://air4thai.pcd.go.th/webV2/download.php) due to many factors contributing to PM 2.5 pollution in this area (e.g., road construction, car repair shops, and small and large factories) (Table 5). The summary for statistics were n = 31, μ ˆ=56.6161,λ ˆ=225.1443, and CV = 0.5015. A Q–Q plot of these data determining the appropriateness of using an IG distribution is shown in Fig. 4. Furthermore, it was found that the AIC and BIC values for the IG distributions were lower than other tested distributions (Table 6), and thus it provided the best fit for the data. The 95% confidence intervals for the CV of this dataset by using the five methods are reported in Table 7. In agreement with the simulation results for n = 30 the F-HPDCI method provided the best confidence interval performance in terms of coverage probability and average length.

Table 2 Daily PM 2.5 data (µg/m3) in January 2021 of Din Daeng district, Bangkok, Thailand.

Day	1	2	3	4	5	6	7	8	9	10	11	
PM 2.5	27.54	32.58	47.29	54.88	42.25	47.46	37.83	29.88	23.58	32.63	35.75	
Day	12	13	14	15	16	17	18	19	20	21	22	
PM 2.5	30.00	52.92	77.00	93.46	105.79	60.46	34.63	53.96	77.79	82.08	91.88	
Day	23	24	25	26	27	28	29	30	31			
PM 2.5	89.71	47.96	44.83	41.33	49.46	51.00	49.04	51.58	50.26			
Notes.

Source: Air Quality and Noise Management Bureau, Pollution Control Department, Thailand.

http://air4thai.pcd.go.th/webV2/download.php.

Figure 3 The inverse Gaussian Q–Q plot for the particulate matter data in Din Daeng district, Bangkok Thailand.

Table 3 AIC and BIC results to check the distribution of PM 2.5 data from Din Daeng district, Bangkok, Thailand.

Densities	Normal	Lognormal	Cauchy	Exponential	Weibull	Inverse Gaussian	
AIC	282.9754	275.0629	286.3146	310.5793	280.4643	274.7625	
BIC	285.8434	277.9309	289.1826	312.0133	283.3322	277.6305	
Notes.

Bold denoted as the minimum AIC and BIC.

Table 4 The 95% confidence intervals for single coefficient of variation of the PM 2.5 data from Din Daeng district, Bangkok, Thailand.

Methods	Confidence intervals for θ	Length of intervals	
	Lower	Upper		
GCI	0.3168	0.5470	0.2302	
AGCI	0.3231	0.5405	0.2174	
BPCI	0.2955	0.4752	0.1797	
FCI	0.3180	0.5426	0.2247	
F-HPDCI	0.3094	0.5289	0.2195	

Table 5 Daily PM 2.5 data (µg/m3) in January 2021 of Bang Khun Thian district, Bangkok, Thailand.

Day	1	2	3	4	5	6	7	8	9	10	11	
PM 2.5	24.00	29.42	44.08	45.64	67.38	84.54	62.17	25.63	19.04	35.29	37.14	
Day	12	13	14	15	16	17	18	19	20	21	22	
PM 2.5	35.71	62.58	99.38	112.08	73.70	71.67	33.27	58.79	93.58	100.33	130.67	
Day	23	24	25	26	27	28	29	30	31			
PM 2.5	76.83	36.38	36.29	37.63	37.00	42.25	39.17	53.50	49.96			
Notes.

Source: Air Quality and Noise Management Bureau, Pollution Control Department, Thailand.

http://air4thai.pcd.go.th/webV2/download.php.

Figure 4 The inverse Gaussian Q–Q plot for the particulate matter data in Bang Khun Thian district, Bangkok Thailand.

Discussion

The results show that the AGCI method performed well in all of the scenarios with large sample sizes as its coverage probabilities were consistently greater than or close to the nominal confidence level while its average lengths were the shortest. In addition, FCI and F-HPDCI produced similar results and performed well for small sample sizes. Moreover, when the sample size and λ increased, the average lengths of all of the methods were reduced. The findings of this work are significantly different from previous related studies because we developed a new method for predicting the air pollution level based on fiducial inference derived by using a Gibbs sampler. Moreover, the proposed methods provided the narrowest average lengths, and so can be used to effectively and accurately estimate confidence intervals for various distributions in various fields. Our approach could aid environmentalists and policymakers to monitor air pollution in specific locations and give alarm signals when the air pollution level approaches a dangerous level. Moreover, the proposed method can also be used in air pollution monitoring systems to mitigate the damage caused by climate change and poor air quality. In the same way, the authorities could leverage our research results to control environmental, social, and health impacts by promoting laws and regulations to ban vehicles with black smoke emission and or diesel engines, forbid people from burning rubbish, develop integrated urban planning with emission reduction policies, replace delivery trucks with electric vehicles, and collect environmental tax or fees according to the “polluter pays” principle.

Table 6 AIC and BIC results to check the distribution of PM 2.5 data from Bang Khun Thian district, Bangkok, Thailand.

Densities	Normal	Lognormal	Cauchy	Exponential	Weibull	Inverse Gaussian	
AIC	298.1317	289.2471	306.2360	314.2502	293.3355	288.7999	
BIC	300.9997	292.1151	309.1040	315.6842	296.2035	291.6679	
Notes.

Bold denoted as the minimum AIC and BIC.

Table 7 The 95% confidence intervals for single coefficient of variation of the PM 2.5 data from Bang Khun Thian district, Bangkok, Thailand.

Methods	Confidence intervals for θ	Length of intervals	
	Lower	Upper		
GCI	0.3965	0.7022	0.3058	
AGCI	0.4068	0.6781	0.2713	
BPCI	0.3788	0.6013	0.2225	
FCI	0.4001	0.6943	0.2942	
F-HPDCI	0.3833	0.6686	0.2853	

Conclusions

In summary, this research paper aims to propose the construction of confidence intervals for the CV of an IG distribution by using the GCI, AGCI, BPCI, FCI, and F-HPDCI approaches. The performances of these methods were compared using the coverage probability and the average length via simulations studies. The results show that AGCI and FCI performed the best in situations with large (n = 50, 100,  and 200) and small (n = 5, 10 and 30) samples sizes, respectively, and thus, they can be recommended for constructing confidence intervals for the CV of an IG distribution in these two scenarios. Finally, two real pollution datasets were utilized to analyze the performance of the proposed method in real situations. In future work, carbon monoxide, lead, nitrogen dioxide, ozone, sulfur dioxide, and other criteria pollutants should be examined by the CV of IG distribution. Furthermore, new credible intervals based on Bayesian inference for the CV of an IG distribution could be developed.

Supplemental Information

Data S1 Daily PM 2.5 data of Din Daeng district in January 2021

Click here for additional data file.

Supplemental Information 2 R code for computing coverage probability and average length

Click here for additional data file.

Dat S2 Daily PM 2.5 data of Bang Khun Thian district in January 2021

Click here for additional data file.

Additional Information and Declarations

Competing Interests

Author Contributions

Data Availability

The authors declare there are no competing interests.

Wasana Chankham conceived and designed the experiments, performed the experiments, analyzed the data, prepared figures and/or tables, authored or reviewed drafts of the paper, and approved the final draft.

Sa-Aat Niwitpong conceived and designed the experiments, authored or reviewed drafts of the paper, and approved the final draft.

Suparat Niwitpong performed the experiments, prepared figures and/or tables, authored or reviewed drafts of the paper, and approved the final draft.

The following information was supplied regarding data availability:

Data set and R code are available in the Supplementary Files.

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
