# Peer review of "Measurement of dispersion of PM 2.5 in Thailand using confidence intervals for the coefficient of variation of an inverse Gaussian distribution"

_PeerJ, doi:10.7717/peerj.12988_

## Round 0.1 · original submission · Major Revisions

There are several shortcomings pointed out by the reviewers, such as the following two concerns.

The “dispersion of PM 2.5” appears in the title. However, the measurement data are used as a test data set in the paper without further discussion. As suggested by reviewer 1, the authors need to show the relevance of proposed techniques to the mentioned air pollution studies. How such studies will benefit the environmental applications is not clear and requires explanations.

Reviewer 2 questioned the originality of the paper. How different is this manuscript from the other paper the authors published, “Confidence Intervals for Coefficient of Variation of Inverse Gaussian Distribution" published on ICVISP 2019?

Please address all the concerns raised by the reviewers.

Reviewer 1 ·

Basic reporting

The authors try to study the PM2.5 time-series in Thailand using confidence intervals for the coefficient of variation of an inverse Gaussian distribution, seems to be an interesting idea but requires more work to be fruitful, so my concerns are as bellow:
Introduction:
1) Page 2, lines 47-55: Could you please bring their results and discuss them and show the relevance of proposed techniques to the mentioned air pollution studies? To me as an environmental researcher, it is a little bit confusing and difficult to follow.
Materials and methods:
1) Please provide the source of air quality data as a separate sub-section.
Results and discussion:
1) Could you please explain in more detail why you chose n, µ, and the number of replicants as mentioned?
2) Please also improve the quality of the Figures, they are hard to read.
3) Please explain in more detail, how your results can help an environmentalist or policymakers?
4) What are you trying to explain which can be interesting in terms of the above-mentioned statements related to air pollution focusing on PM2.5 and its environmental, social, and health impacts?
5) The discussion is not well provided and requires to be re-written, and the final aim of the work is not clear, please bring more related studies to compare your results and show how important and/or useful your findings or proposed metho are.
Conclusion:
1) The conclusion is a repeat of the abstract and does not provide new information, please re-write it completely!
2) Please talk more in detail about the future works and again how they can add more value to the existing knowledge!

Experimental design

1) Please provide the source of air quality data as a separate sub-section.

Validity of the findings

1) Could you please explain in more detail why you chose n, µ, and the number of replicants as mentioned?
2) Please also improve the quality of the Figures, they are hard to read.
3) Please explain in more detail, how your results can help an environmentalist or policymakers?
4) What are you trying to explain which can be interesting in terms of the above-mentioned statements related to air pollution focusing on PM2.5 and its environmental, social, and health impacts?
5) The discussion is not well provided and requires to be re-written, and the final aim of the work is not clear, please bring more related studies to compare your results and show how important and/or useful your findings or proposed metho are.

Annotated reviews are not available for download in order to protect the identity of reviewers who chose to remain anonymous.

Reviewer 2 ·

Basic reporting

No comment

Experimental design

No comment

Validity of the findings

No comment

Additional comments

In general in terms of writing and structure, I think that the manuscript has its goal clearly stated, along with sufficient details and information in method description.

However, in term of original research, I would like to ask the authors to clarify on the difference between the current manuscript and a previous manuscript by the same authors. Namely, "Confidence Intervals for Coefficient of Variation of Inverse Gaussian Distribution" published on ICVISP 2019: Proceedings of the 3rd International Conference on Vision, Image and Signal Processing (August 2019 Article No.: 73Pages 1–6. https://doi.org/10.1145/3387168.3387254). Two of methods in the current manuscript are the same as those in the above-mentioned previously published manuscript.

---

## Round 0.2 · accepted · Accept

Thanks for addressing all the reviewers' concerns in your revised manuscript.

Reviewer 1 ·

Basic reporting

No further comments!

Experimental design

No

Validity of the findings

No comment

Additional comments

Thank you for addressing all comments!